# MULTIMODAL STRUCTURE PRESERVATION LEARNING

## ABSTRACT

When selecting data to build machine learning models in practical applications, factors such as availability, acquisition cost, and discriminatory power are crucial considerations. Different data modalities often capture unique aspects of the underlying phenomenon, making their utilities complementary. On the other hand, some sources of data host structural information that is key to their value. Hence, the utility of one data type can sometimes be enhanced by matching the structure of another. We propose Multimodal Structure Preservation Learning (MSPL) as a novel method of learning data representations that leverages the clustering structure provided by one data modality to enhance the utility of data from another modality. We demonstrate the effectiveness of MSPL in uncovering latent structures in synthetic time series data and recovering clusters from whole genome sequencing and antimicrobial resistance data using mass spectrometry data in support of epidemiology applications. The results show that MSPL can imbue the learned features with external structures and help reap the beneficial synergies occurring across disparate data modalities.

## 1 INTRODUCTION

Selecting the appropriate data is critical when deploying machine learning models in real-world applications. Factors such as availability, acquisition cost, and discriminatory power (reflected by information density, resolution, etc.) are of primary concern. On the other hand, distinct data modalities may encode different information about the same underlying phenomenon (Xu et al., 2013; Baltrušaitis et al., 2018), thus forming a gap in utility. For instance, in medical diagnostics, imaging data such as X-ray and CT scans reveal structural anomalies, while genomic sequencing offers insights into the molecular mechanisms of diseases (Esteva et al., 2019).

Traditionally, research in multimodal machine learning attempts to bridge this gap by learning a shared feature space between data modalities (Li et al., 2023; Liu et al., 2024b;a). Deviating from these feature-level alignment approaches that require complete data in two modalities, we approach the problem from *structure*-level alignment: In scenarios such as clustering, it is the structure of the data that directly influences the results. Thus, learning representations of one data modality that preserve the structure of another can extend the utility of the former and effectively bridge their gap.

For instance, in hospital outbreak investigations, epidemiologists use the single nucleotide polymorphism (SNP) distance defined on whole genome sequencing (WGS) data to cluster microbial samples, assess their lineages, and identify outbreaks. WGS provides the highest discriminatory power and is considered the gold standard in microbial disease epidemiology (Bertelli & Greub, 2013). However, the labor, cost, and expertise required for WGS make it prohibitive to deploy broadly (Rossen et al., 2018). On the other hand, due to its low cost and rapid time to generate results, Matrix-Assisted Laser Desorption Ionization–Time of Flight (MALDI-TOF) mass spectrometry rose as a standard tool for microbial species identification in clinical microbiology laboratories (Croxatto et al., 2012; Clark et al., 2013). Though MALDI has weaker discriminatory power than WGS, it is gaining attention as a potential cost-effective alternative to WGS for hospital outbreak detection (Griffin et al., 2012). Hence, if MALDI representations that preserve the SNP distance structure can be learned, its utility can extend from species identification to outbreak detection, making it a viable substitute for WGS in practice.

To achieve the aforementioned goals, we propose a novel machine learning framework called **Multimodal Structure Preservation Learning (MSPL)** that learns data representations that enhance

the utility of one data modality through alignment with the dissimilarity-based clustering structure provided by another data modality. We first demonstrate the effectiveness of MSPL in identifying latent structures on a synthetic time series dataset. We then apply MSPL to epidemiology settings, where we enhance the utility of MALDI mass spectrometry by leveraging the clustering structure in whole genome sequencing (WGS) and antimicrobial resistance (AMR) data, respectively. Our results demonstrate that MSPL can effectively inject structural information of one modality into the representations of another, improve the clustering performance, bridge the utility gap of two modalities, substantially reduce data acquisition cost, and increase feasibility of learning.

## 2 RELATED WORK

**Multimodal connectors**. In multimodal machine learning models, connectors are employed to bridge the gap between different modalities in the feature space. In vision-language pre-training, Li et al. (2023) proposed the Querying Transformer (Q-Former), a connector that learns query vectors to extract the visual features most relevant to the text. The Q-Former architecture has also been adopted to align time series and text features (Cai et al., 2023). Llava (Liu et al., 2024b), a pioneering work in visual instruction fine-tuning, introduced a more lightweight connector using just a linear projection that projects image features onto the word embedding space. Liu et al. (2024a) later improved Llava's multimodal capabilities by changing the linear projection connector to a two-layer multilayer perceptron (MLP), which affords more representation power. However, Qi et al. (2024) found that multimodal connectors can be performance bottlenecks for multimodal large language models and may fall short with insufficient training data compared to the amount of pre-training data. They proposed to enhance the connector with retrieval-augmented tag tokens that contain rich object-aware information.

**Structure in multimodal self-supervised learning**. Pre-training approaches in multimodal self-supervised learning, e.g., CLIP (Radford et al., 2021), commonly employ an instance-wise contrastive objective (Oord et al., 2018) to learn joint features. However, the contrastive objective ignores the underlying semantic structure across samples (Zellers et al., 2021; Singh et al., 2022), and thus may adversely impact model performance. To remedy this issue, Chen et al. (2021) proposed to combine the contrastive objective with a joint multimodal clustering objective to capture the cross-modal semantic similarity structure. Alternatively, Swetha et al. (2023) preserved modality-specific relationships in the joint embedding space by learning semantically meaningful "anchors" and representing inter-sample relations with sample-anchor relations.

**Deep learning for MALDI spectrometry**. Deep learning is widely applied to MALDI spectra analysis. Weis et al. (2022) used an MLP to encode MALDI spectra of bacterial strains and predict their antimicrobial resistance to a range of drugs. Normand et al. (2022) utilized a 1-D CNN to identify a subpopulation from MALDI data of the same species. Abdelmoula et al. (2021) employed a variational autoencoder to learn low-dimensional latent features of MALDI spectra that reveal biologically relevant clusters of tumor regions.

## 3 METHODS

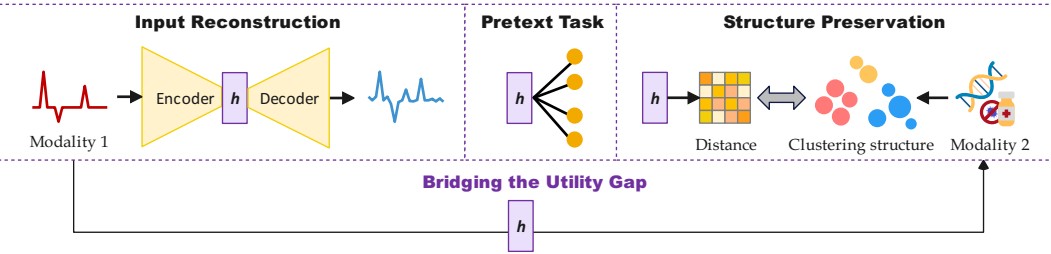

Figure 1: Overview of the MSPL framework.

**Multimodal Structure Preservation Learning.** Figure 1 presents an overview of the MSPL framework. MSPL entails three objectives: (1) reconstruction of the input data as in standard au-

toencoders for feature extraction, (2) a pretext task on which the input data has discriminatory power, and (3) structure preservation through alignment between the clustering structure of two modalities.

Formally, the input data to MSPL is $\boldsymbol{x}$ with batch size $N$. An autoencoder consisting of an encoder $\mathrm{Enc}_x(\cdot)$ and a decoder $\mathrm{Dec}(\cdot)$ encodes $\boldsymbol{x}$ to a latent representation $\boldsymbol{h}_0$ and uses it to obtain the reconstructed input data $\hat{\boldsymbol{x}}$. A further encoding step $\mathrm{Enc}_h(\cdot)$ prepares $\boldsymbol{h}_0$ for the pretext task and structure preservation. In all our experiments, the autoencoder adopts the U-Net (Ronneberger et al., 2015) architecture on 1-D data, and the pretext task is defined as a classification task, accomplished by the classification head $\mathrm{CLS}(\cdot)$. The mathematical formulation for the MSPL framework is as follows:

$$\boldsymbol{h}_0 = \mathrm{Enc}_x(\boldsymbol{x}), \tag{1}$$

$$\hat{\boldsymbol{x}} = \mathrm{Dec}(\boldsymbol{h}_0), \tag{2}$$

$$\boldsymbol{h} = \mathrm{Enc}_h(\boldsymbol{h}_0), \tag{3}$$

$$\boldsymbol{z} = \mathrm{CLS}(\boldsymbol{h}). \tag{4}$$

We then define the loss functions corresponding to the three objectives:

$$\mathcal{L}_{\mathrm{recon}} = \frac{1}{N}\|\boldsymbol{x} - \hat{\boldsymbol{x}}\|_2^2, \tag{5}$$

$$\mathcal{L}_{\mathrm{pretext}} = \mathrm{CE}(\boldsymbol{z}, \boldsymbol{y}), \tag{6}$$

$$\mathcal{L}_{\mathrm{struct}} = f_{\mathrm{struct}}(\mathrm{pdist}(\boldsymbol{h}), \boldsymbol{d}). \tag{7}$$

Here, $\boldsymbol{y}$ represents the labels for the pretext task, $\mathrm{CE}(\cdot)$ stands for the cross-entropy loss, $\mathrm{pdist}(\boldsymbol{h})$ computes the $\ell_2$ distance between each pair of row vectors in the learned feature matrix $\boldsymbol{h}$, and $\boldsymbol{d}$ refers to the external dissimilarity matrix computed from another modality. The function $f_{\mathrm{struct}}$ matches these dissimilarities measured in the two modalities and by default is implemented as the mean squared error, but can vary as required by application.

The loss for MSPL is a weighted sum of these three losses:

$$\mathcal{L}_{\mathrm{MSPL}} = \mathcal{L}_{\mathrm{recon}} + \lambda_0 \mathcal{L}_{\mathrm{pretext}} + \lambda_1 \mathcal{L}_{\mathrm{struct}}, \tag{8}$$

where $\lambda_0$ and $\lambda_1$ are hyperparameters controlling the relative weights of the component losses.

Through MSPL, we aim to learn input data representations $\boldsymbol{h}$ that retain discriminatory power on the pretext task while preserving the clustering structure characterized by the external dissimilarity matrix $\boldsymbol{d}$. For example, we learn representations of MALDI spectra that readily identify species (a pretext task) while their pairwise distances recover the clusters defined by the dissimilarity among WGS or AMR data.

**Baseline models.** In comparison with MSPL, we develop two baseline models. The first model, named "onlyCLS," constructs an ablation study by removing the structure preservation objective, i.e., $\mathcal{L}_{\mathrm{struct}}$, from the MSPL loss. Hence, onlyCLS can only rely on the feature extraction capabilities of the autoencoder and the discriminatory power of the pretext task to implicitly recover the clustering structure of another modality.

The second model, named "clusCLS," formulates structure preservation through classification. Specifically, the ground truth cluster labels $\mathcal{C}_T$ are derived beforehand using the external dissimilarity matrix $\boldsymbol{d}$ on the full dataset. Then, an additional classification head $\mathrm{CLS}_C(\cdot)$ is used to classify these cluster labels:

$$\boldsymbol{z}_c = \mathrm{CLS}_C(\boldsymbol{h}\|\boldsymbol{z}), \tag{9}$$

where "$\|$" stands for the concatenation operation, and $\boldsymbol{h}$ and $\boldsymbol{z}$ are the learned features and pretext classification logits as computed in Eqs. 3 and 4, respectively. clusCLS also replaces $\mathcal{L}_{\mathrm{struct}}$ with the following cross-entropy loss:

$$\mathcal{L}_{\mathrm{structCLS}} = \mathrm{CE}(\boldsymbol{z}_c, \mathcal{C}_T). \tag{10}$$

**Cluster evaluation.** We cluster the learned representations $\boldsymbol{h}$ and measure the similarity of the generated clusters to those derived from the external dissimilarity matrix $\boldsymbol{d}$. Besides adopting the Adjusted Rand Index (ARI) and Normalized Mutual Information (NMI) as standard evaluation metrics, we propose an alternative cluster similarity metric, *cluster F1 score*, as described below.

We first define the *purity* of a cluster assignment with respect to ground truth labels: given a dataset $X$, a set of clusters $\{C_1, \cdots, C_p\}$ and a label set $Y$ on $X$, the purity score for the $j$-th cluster is defined as

$$\text{Purity}(C_j, Y) = \frac{1}{|C_j|} \max_{k \in Y} |C_j \cap T_k|, \tag{11}$$

where $T_k$ is the set of data points in $X$ with label $k$.

We then extend the purity metric to our cluster evaluation setting. Specifically, given dataset $X$, predicted clusters $\mathcal{C}_S = \{C_S^1, \cdots, C_S^p\}$ derived from the input data representations $\boldsymbol{h}$, and "ground truth" clusters $\mathcal{C}_T = \{C_T^1, \cdots, C_T^q\}$ derived from the external dissimilarity matrix $\boldsymbol{d}$, we define the cluster *precision*, *recall*, and *F1 score* of the predicted $\mathcal{C}_S$ with respect to $\mathcal{C}_T$ as follows:

$$\text{Prec}(\mathcal{C}_S, \mathcal{C}_T) = \frac{1}{p} \sum_{i=1}^{p} \text{Purity}(C_S^i, \mathcal{C}_T), \tag{12}$$

$$\text{Rec}(\mathcal{C}_S, \mathcal{C}_T) = \frac{1}{q} \sum_{j=1}^{q} \text{Purity}(C_T^j, \mathcal{C}_S), \tag{13}$$

$$\text{F1}(\mathcal{C}_S, \mathcal{C}_T) = 2 \cdot \frac{\text{Prec}(\mathcal{C}_S, \mathcal{C}_T) \cdot \text{Rec}(\mathcal{C}_S, \mathcal{C}_T)}{\text{Prec}(\mathcal{C}_S, \mathcal{C}_T) + \text{Rec}(\mathcal{C}_S, \mathcal{C}_T)}. \tag{14}$$

To reach high precision, each predicted cluster must contain as few distinct ground truth cluster labels as possible (i.e., be pure). To achieve high recall, the data points with the same ground truth label should be clustered together in the predictions. The defined precision reaches its maximum value 1 when $p = |X|$ and $|C_S^i| = 1$ for all $i$, while the recall reaches its maximum value 1 when $p = 1$ and $|C_S^1| = |X|$. In contrast, a cluster assignment with a high $F_1$ score does not fall into either extreme. Nevertheless, to avoid the impact of singleton ground truth clusters ($|C_T^j| = 1$) on the above metrics, we only evaluate them on the subset of the dataset where the ground truth cluster has more than 1 element, i.e., $\cup_{j:|C_T^j| \geq 2} C_T^j \subseteq X$.

## 4 DATASETS

To demonstrate the ability of MSPL to learn representations that preserve external clustering structure from another modality, we utilized three datasets: a synthetic time series dataset (Synth-TS), a proprietary dataset of MALDI spectra with paired whole genome sequencing SNP distance profiles, and a public dataset of MALDI spectra paired with AMR profiles, Database of Resistance Information on Antimicrobials and MALDI-TOF Mass Spectra (DRIAMS) (Weis et al., 2022).

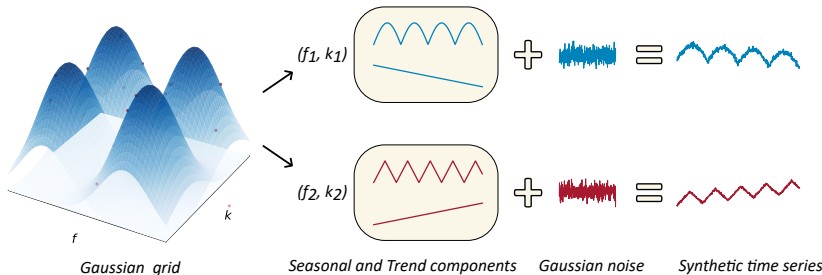

Figure 2: Generating the Synth-TS dataset.

**Synth-TS.** We construct a synthetic time series dataset called Synth-TS to demonstrate the ability of MSPL to learn representations that preserve external structure. Each time series in Synth-TS is a superposition of three components: (1) a seasonal component consisting of full-wave rectified sine waves (i.e., the absolute value of a sine wave) or triangle waves, (2) a trend component consisting of a linearly increasing or declining trend, and (3) a Gaussian noise component. Each time series is parameterized by the frequency $f$ of the seasonal component and the slope $k$ of the trend component.

As shown in Figure 2, $(f, k)$ are jointly sampled from a two-dimensional Gaussian distribution. In Synth-TS, we construct a grid of such two-dimensional Gaussians and randomly generate multiple time series from each Gaussian. Formally, the dataset Synth-TS$(m, n)$ generates $2n$ samples from each Gaussian in a $m \times m$ grid, with $n$ samples having a sine wave component and $n$ samples having a triangle wave component.

We define the pretext task as the binary classification of seasonal components (sine or triangle waves). The MSPL framework learns representations of the time series such that their pairwise distances match the external pairwise dissimilarity, which we defined as the Euclidean distance between their respective parameters $(f, k)$. This encourages the time series to cluster according to the Gaussian distributions generating their parameters. More details about the construction of Synth-TS can be found in Appendix A.

**Proprietary dataset.** The proprietary dataset consists of $1862$ bacterial samples with MALDI spectra spanning $42$ species with corresponding WGS information from a single hospital. Though the raw WGS data were unavailable, we have access to the pairwise dissimilarity of this data, measured by SNP distance. In our framework, species identification from MALDI spectra forms the pretext classification task, and the SNP distances define the external dissimilarity matrix $\boldsymbol{d}$ (Eq. 7).

In bacterial outbreak investigations, SNP distances lower than a pre-defined threshold indicate closely related or nearly identical strains that may form an outbreak cluster (Guerra-Assunção et al., 2015; Hatherell et al., 2016). In our experiments, the threshold is set to $15$ as used by epidemiologists (Xiao et al., 2024). To derive the ground truth outbreak clusters from the SNP distances, we apply hierarchical clustering with complete linkage on the full SNP distance matrix, using $15$ as the distance threshold.

While small SNP distances are crucial to outbreak cluster detection, the distances can vary drastically in scale, ranging from less than $10$ to more than $10^5$. Here, we develop a custom loss function for the structure preservation objective that avoids potential overfitting to large SNP distances:

$$\mathcal{L}_{\text{struct}} = \frac{1}{N^2} \sum_{i,j \in [N]^2} f_{\text{SNP}}(\text{pdist}(\boldsymbol{h})_{ij}, \boldsymbol{d}_{ij}, t), \tag{15}$$

$$f_{\text{SNP}}(x, y, t) = \begin{cases} (x - y)^2 & y \leq t, \\ (\max\{0, t - x\})^2 & y > t, \end{cases} \tag{16}$$

where $\text{pdist}(\boldsymbol{h})$ and $\boldsymbol{d}$ are the feature distance and SNP distance matrices, respectively (Eq. 7), $N$ is the batch size, and $t$ is the chosen SNP threshold. Under this custom loss function, no penalty is imposed when both the feature distance and SNP distance exceed the SNP threshold, as they have no impact on the practice of outbreak detection.

**DRIAMS data.** DRIAMS (Weis et al., 2022) is a public dataset with paired MALDI spectra and antimicrobial resistance (AMR) profiles. In our experiments, we use the DRIAMS-B and DRIAMS-C subsets, which consist of $10404$ bacterial samples with MALDI spectra spanning $251$ species. As above, we define species identification from MALDI spectra as the pretext classification task.

The AMR profile of each bacterial sample documents its resistance to various antibiotics. We utilize the AMR profiles against $33$ shared drugs recorded in both DRIAMS subsets. We preprocess the AMR profiles to construct the external dissimilarity matrix $\boldsymbol{d}$, where each entry is an integer ranging from $0$ to $33$. Details about the AMR profile preprocessing can be found in Appendix B.

To derive ground truth outbreak clusters from the AMR dissimilarity matrix, we apply hierarchical clustering with varying distance thresholds from $1$ to $33$. The optimal threshold—chosen to maximize the number of non-singleton clusters—is set to $10$ for ground truth generation.

## 5 EXPERIMENTS AND RESULTS

In our experiments, we adopt two different clustering schemes for the features $\boldsymbol{h}$ learned from MSPL and onlyCLS. The first scheme involves hierarchical clustering with a distance threshold, denoted by

the subscript "thr." For the proprietary dataset and DRIAMS, we first determine the distance threshold that yields the optimal cluster F1 score on the training data. Specifically, for the proprietary dataset, the upper bound of the thresholds is set to 20, an alternative threshold used by epidemiologists for outbreak detection (Szarvas et al., 2021), which is close to the threshold we used to obtain groud-truth clusters. For DRIAMS, the upper bound is set to 33, the maximum dissimilarity between AMR profiles (see Appendix B). The same threshold is then used for evaluation. The second scheme also employs hierarchical clustering, but the number of output clusters is set to match that of the ground truth. We denote this scheme by the subscript "num." For Synth-TS, only the second clustering scheme is used since the ground truth clusters are not derived from hierarchical clustering with a distance threshold.

For evaluation, we perform 2-fold cross-validation on Synth-TS and the proprietary dataset and 5-fold cross-validation for DRIAMS. Each cross-validation experiment is repeated over 5 random trials, using different random seeds for data splitting. We first average the metrics across the validation folds, then report the mean and the $95\%$ confidence interval (using $t$-distribution) of the averaged metrics across the 5 random trials.

| Dataset | Model | ARI | NMI | Precision | Recall | F1 Score | Pretext Accuracy |
|---------|-------|-----|-----|-----------|--------|----------|------------------|
| Synth-TS$(5, 80)$ | MSPL$_{num}$ | $0.426 \pm 0.011$ | $\mathbf{0.734 \pm 0.003}$ | $0.609 \pm 0.019$ | $0.611 \pm 0.018$ | $0.61 \pm 0.019$ | $0.984 \pm 0.002$ |
| | onlyCLS$_{num}$ | $0.175 \pm 0.009$ | $0.581 \pm 0.008$ | $0.509 \pm 0.014$ | $0.584 \pm 0.015$ | $0.543 \pm 0.013$ | $0.986 \pm 0.002$ |
| | clusCLS | $\mathbf{0.462 \pm 0.008}$ | $0.727 \pm 0.003$ | $\mathbf{0.667 \pm 0.01}$ | $\mathbf{0.662 \pm 0.008}$ | $\mathbf{0.664 \pm 0.009}$ | $0.976 \pm 0.003$ |
| Synth-TS$(10, 20)$ | MSPL$_{num}$ | $\mathbf{0.395 \pm 0.007}$ | $\mathbf{0.819 \pm 0.001}$ | $0.605 \pm 0.005$ | $\mathbf{0.6 \pm 0.005}$ | $\mathbf{0.602 \pm 0.004}$ | $0.982 \pm 0.002$ |
| | onlyCLS$_{num}$ | $0.169 \pm 0.019$ | $0.708 \pm 0.012$ | $\mathbf{0.609 \pm 0.024}$ | $0.515 \pm 0.023$ | $0.558 \pm 0.021$ | $0.987 \pm 0.002$ |
| | clusCLS | $0.32 \pm 0.014$ | $0.78 \pm 0.005$ | $0.561 \pm 0.007$ | $0.556 \pm 0.008$ | $0.559 \pm 0.007$ | $0.982 \pm 0.002$ |
| Synth-TS$(16, 10)$ | MSPL$_{num}$ | $\mathbf{0.386 \pm 0.005}$ | $\mathbf{0.869 \pm 0.001}$ | $0.635 \pm 0.007$ | $\mathbf{0.631 \pm 0.005}$ | $\mathbf{0.633 \pm 0.006}$ | $0.988 \pm 0.002$ |
| | onlyCLS$_{num}$ | $0.179 \pm 0.018$ | $0.798 \pm 0.007$ | $\mathbf{0.668 \pm 0.011}$ | $0.541 \pm 0.009$ | $0.598 \pm 0.008$ | $0.991 \pm 0.001$ |
| | clusCLS | $0.234 \pm 0.004$ | $0.821 \pm 0.001$ | $0.552 \pm 0.006$ | $0.53 \pm 0.006$ | $0.541 \pm 0.006$ | $0.979 \pm 0.003$ |
| Proprietary Dataset | MSPL$_{thr}$ | $0.001 \pm 0.002$ | $0.137 \pm 0.04$ | $0.962 \pm 0.021$ | $\mathbf{0.962 \pm 0.009}$ | $\mathbf{0.962 \pm 0.011}$ | $0.813 \pm 0.029$ |
| | MSPL$_{num}$ | $0.034 \pm 0.001$ | $0.884 \pm 0.001$ | $0.935 \pm 0.007$ | $0.605 \pm 0.009$ | $0.734 \pm 0.008$ | |
| | onlyCLS$_{thr}$ | $0.0 \pm 0.004$ | $\mathbf{0.897 \pm 0.01}$ | $\mathbf{0.986 \pm 0.004}$ | $0.525 \pm 0.012$ | $0.685 \pm 0.009$ | $0.864 \pm 0.007$ |
| | onlyCLS$_{num}$ | $0.030 \pm 0.003$ | $0.88 \pm 0.001$ | $0.937 \pm 0.007$ | $0.584 \pm 0.004$ | $0.719 \pm 0.004$ | |
| | clusCLS | $\mathbf{0.437 \pm 0.049}$ | $0.759 \pm 0.015$ | $0.722 \pm 0.018$ | $0.621 \pm 0.016$ | $0.667 \pm 0.006$ | $0.804 \pm 0.003$ |
| DRIAMS | MSPL$_{thr}$ | $0.207 \pm 0.074$ | $0.54 \pm 0.058$ | $\mathbf{0.984 \pm 0.004}$ | $\mathbf{0.896 \pm 0.016}$ | $\mathbf{0.937 \pm 0.01}$ | $0.912 \pm 0.003$ |
| | MSPL$_{num}$ | $0.005 \pm 0.0001$ | $\mathbf{0.736 \pm 0.001}$ | $0.966 \pm 0.001$ | $0.31 \pm 0.022$ | $0.468 \pm 0.025$ | |
| | onlyCLS$_{thr}$ | $0.130 \pm 0.028$ | $0.665 \pm 0.018$ | $0.976 \pm 0.004$ | $0.716 \pm 0.072$ | $0.818 \pm 0.053$ | $0.966 \pm 0.001$ |
| | onlyCLS$_{num}$ | $0.005 \pm 0.001$ | $0.729 \pm 0.0005$ | $0.970 \pm 0.002$ | $0.302 \pm 0.012$ | $0.46 \pm 0.013$ | |
| | clusCLS | $\mathbf{0.574 \pm 0.026}$ | $0.647 \pm 0.024$ | $0.819 \pm 0.017$ | $0.837 \pm 0.011$ | $0.826 \pm 0.007$ | $0.953 \pm 0.001$ |

Table 1: Model performance on Synth-TS, the proprietary dataset, and DRIAMS.

OBSERVATION 1: MSPL EFFECTIVELY PRESERVES EXTERNAL STRUCTURE

**Uncovering latent structure in Synth-TS.** We evaluate our models on three versions of the Synth-TS dataset: Synth-TS$(5, 80)$, Synth-TS$(10, 20)$, and Synth-TS$(16, 10)$, containing 80, 20, and 10 samples per type of the seasonal component (sine or triangle) per Gaussian, respectively.

As shown in Table 1, while MSPL falls short of clusCLS in terms of F1 score and ARI on Synth-TS$(5, 80)$, it significantly outperforms clusCLS across *all* clustering metrics on Synth-TS$(10, 20)$ and Synth-TS$(16, 10)$, suggesting that MSPL has a marked advantage over the classification approach on sparser datasets.

Furthermore, for the clusCLS model, all reported clustering metrics——except NMI——exhibit a clear downward trend as the number of samples per class decreases. This is expected, as fewer samples make it more challenging for the cluster label classifier to accurately capture class characteristics. In contrast, the metrics for MSPL remain consistent despite the decreasing number of samples per Gaussian, suggesting that MSPL is more effective at preserving external clustering structures and is robust to cluster sparsity.

Additionally, MSPL consistently outperforms onlyCLS in ARI, NMI, cluster recall, and cluster F1 score, underscoring the importance of dissimilarity matching in $\mathcal{L}_{struct}$ for structure preservation.

**Recovering WGS clusters in the proprietary dataset.**

We first observe that when the number of predicted clusters is constrained to match the ground truth, $MSPL_{num}$ outperforms the other models in both NMI and cluster F1 score. Furthermore, $MSPL_{thr}$ vastly outperforms both $onlyCLS_{thr}$ and clusCLS in terms of cluster recall and F1 score. This demonstrates the superior ability of MSPL to preserve structure in real-world settings, effectively mimicking the decision-making criterion of epidemiologists during outbreak investigations.

Figure 3 presents the bipartite graph showing the correspondence between ground truth WGS clusters and predicted clusters for one of the species, *Klebsiella pneumoniae*, with links indicating matched clusters with at least two samples. Our method manages to group data points of the same ground truth cluster together (high recall), though the precision is lower given the two large MALDI clusters, instead of eight smaller WGS clusters.

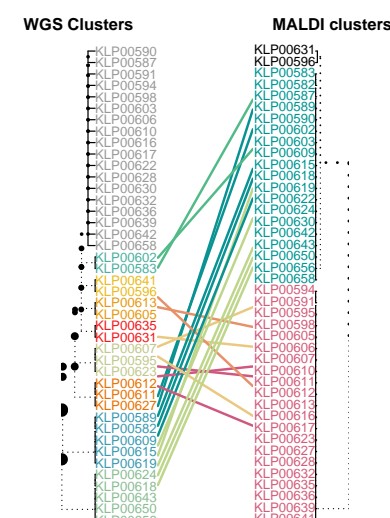

Figure 3: Bipartite graph of *Klebsiella pneumoniae* clusters.

**Recovering AMR clusters in DRIAMS.** As shown in Table 1, MSPL outperforms all baseline models in precision, recall, and F1 score. Additionally, when we constrain the number of output clusters to match the ground truth, $MSPL_{num}$ outperforms other models in NMI. Figure 4 illustrates the ground truth and predicted clusters of data points projected through multidimensional scaling of the predicted dissimilarity matrix ($pdist(\boldsymbol{h})$). These findings further validate MSPL's ability to learn representations that preserve external structure, effectively bridging different modalities.

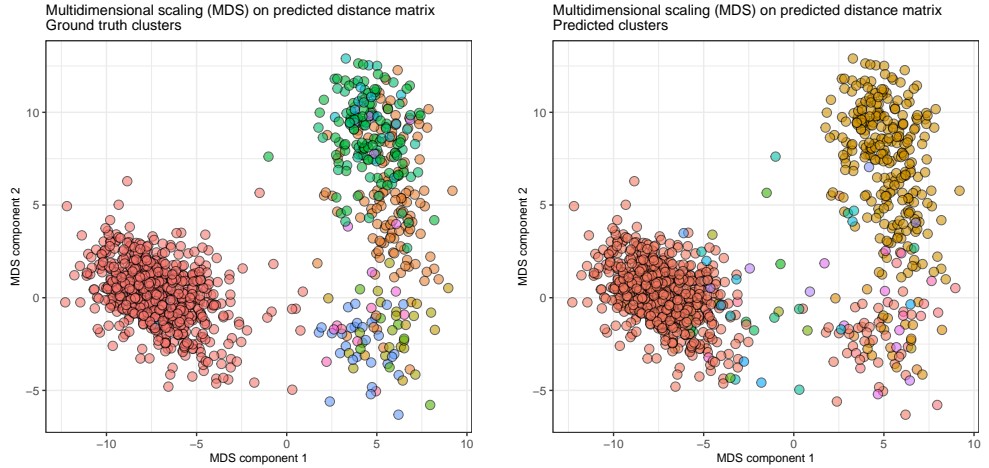

Figure 4: Multidimensional scaling projection based on predicted distance matrix of DRIAMS data using MSPL model, colored in ground truth clusters (left) and predicted clusters (right) respectively. MSPL recovered most of the clusters in ground truth.

OBSERVATION 2: MSPL THRIVES IN DIVERSE DATA SUBSETS

After evaluating the model performance on the full datasets, we now investigate the impact of data substructure on MSPL's performance. In both the proprietary dataset and DRIAMS, the ground truth clusters assigned to MALDI samples within a single bacterial species naturally represent such substructures. For each species-defined subset, we calculate the *lift* in cluster F1-score between

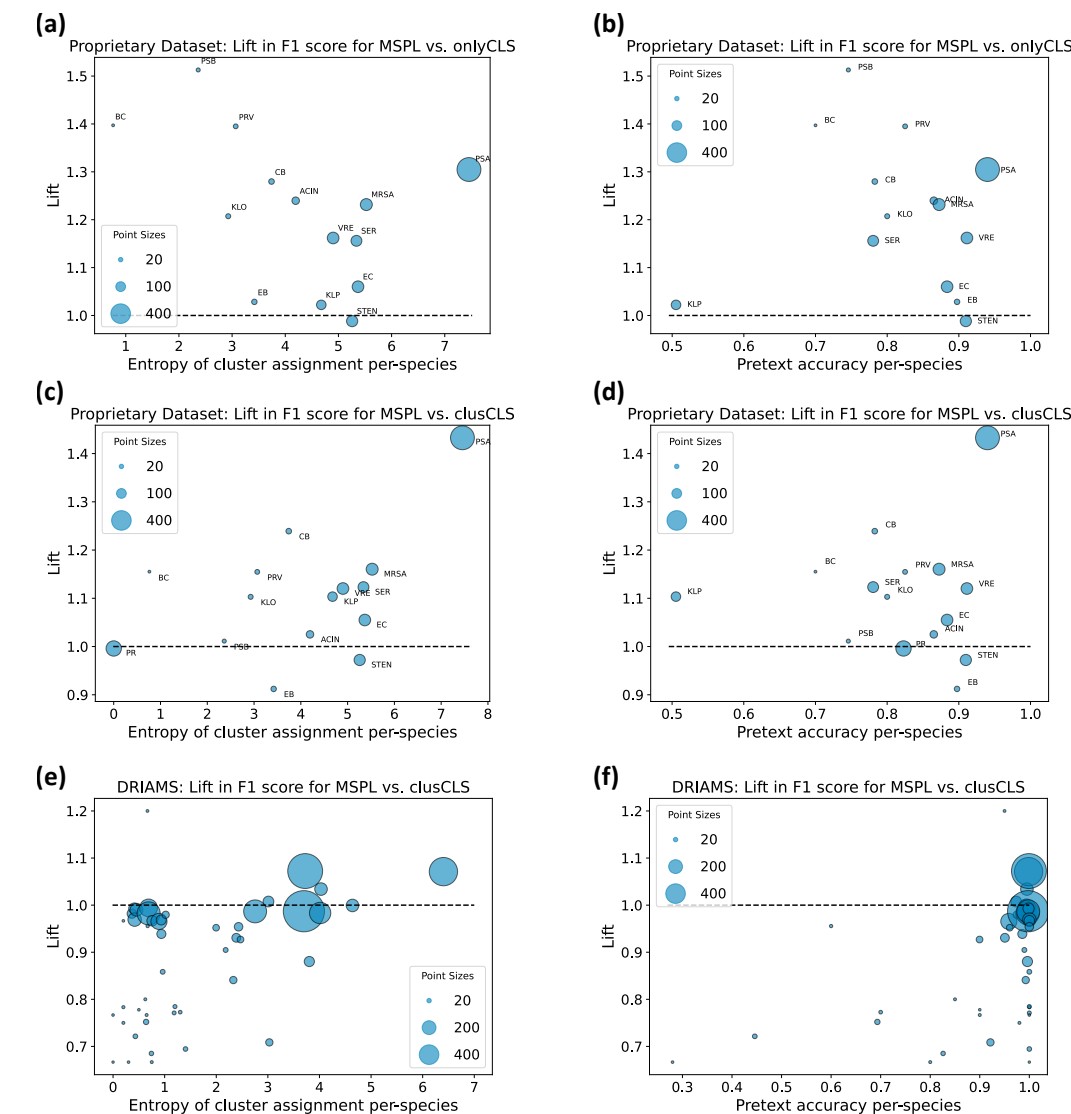

Figure 5: Lift in F1 score for different species in the proprietary dataset and DRIAMS, sorted by the entropy of ground truth clusters or the pretext accuracy. The dot size reflects the number of MALDI samples in that species. Species label descriptions are listed in Appendix C.

MSPL and the two baseline models. The lift is defined as the ratio of MSPL's cluster F1-score to that of clusCLS or onlyCLS. To characterize the substructure, we measure the "diversity" of each subset using the mean Shannon entropy of the ground truth cluster assignment on its samples across the validation sets.

We then plot the lift against the Shannon entropy for all species in both the proprietary dataset and DRIAMS. We find that MSPL achieves outperforms onlyCLS (i.e., lift $> 1$) in all subsets of the proprietary dataset except *Stenotrophomonas maltophilia* (STEN) (Figure 5(a)) and in the majority of subsets in DRIAMS (Appendix D). This result underscores the importance of the dissimilarity matching to MSPL's ability to preserve structure across different subsets of the data.

On the other hand, when compared to clusCLS, MSPL achieves higher lift for species with greater Shannon entropy in both datasets (Figure 5(c,e)). This finding suggests that, compared to the classification approach, MSPL excels when the data exhibits high diversity in cluster distribution.

OBSERVATION 3: MSPL IS ROBUST TO THE DIFFICULTY OF ITS PRETEXT TASK

We further investigate whether the difficulty of the pretext task affects structure preservation in MSPL. Again, we evaluate model performance on the species-defined subsets of both the proprietary dataset and DRIAMS.

To quantify the difficulty of the pretext task for each species, we calculate the mean accuracy of predicting that species across the validation sets. As illustrated in Figure 5(b,d), the relationship between MSPL and baseline F1 scores in the proprietary dataset is agnostic to pretext task accuracy. Notably, the lift can exceed 1 for species with both high and low pretext task accuracies. A similar pattern is observed in DRIAMS (Figure 5(f) and Appendix D). These results suggest that the difficulty of the pretext task does not limit MSPL's ability to preserve structure.

## 6 DISCUSSION

We introduced Multimodal Structure Preservation Learning (MSPL), a method designed to enhance the utility of a data modality by learning representations that preserve external clustering structures from another modality. Through empirical evaluation, we demonstrate that MSPL can effectively capture and preserve latent structures in synthetic time series, whole genome sequencing (WGS), and antimicrobial resistance (AMR) data. Additionally, MSPL offers two advantages: it excels in preserving highly diverse substructures and remains robust to the difficulty of the pretext task. Our approach is a novel addition to the family of multimodal machine learning techniques in that it aggregates information across different forms of its representation, even though the underlying form of data may originally be transactional in both sources. It can be useful in applications where a pattern structure present in some data sources can support models trained on other data sources.

(a)           (b)           (c)

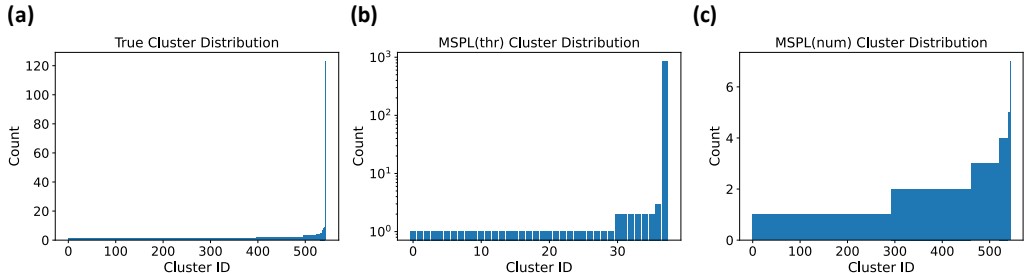

Figure 6: Distributions of clusters on the proprietary dataset. There are $544$ ground truth clusters while MSPL$_{thr}$ predicted only $38$ clusters.

**Limitations.** In our performance evaluation (Table 1), we observed that MSPL yields low NMI scores on both the proprietary dataset and DRIAMS when clustering the learned features using a distance threshold (MSPL$_{thr}$). However, when constraining the number of clusters (MSPL$_{num}$), the NMI score improves and surpasses the baseline, albeit resulting in a drop in the F1 score. Additionally, for both clustering schemes, the ARI remains consistently low. While NMI can be inadequate when comparing predictions to a large number of clusters (Amelio & Pizzuti, 2015), and ARI is more suitable for comparisons involving large, equally sized clusters (Romano et al., 2016), the low NMI and ARI scores indicate a potential limitation of the current MSPL framework in handling imbalanced cluster distributions, as observed in the proprietary dataset (Figure 6). Moreover, in real-world clinical applications (e.g., outbreak detection), the number of clusters is often unknown, making it infeasible to apply the MSPL$_{num}$ model.

**Future work.** Given the limitations of the current MSPL framework, future research will focus on improving it in the face of cluster imbalance. Specifically, we plan to enhance the structure preservation objective ($\mathcal{L}_{struct}$) by incorporating additional supervision that encourages MSPL to accurately estimate the number of ground truth clusters as well as their distribution. We will also experiment with more than one source of structural information (e.g., including WGS *and* AMR simultaneously in support of learning on MALDI data), and consider other than clustering structures as sources of predictive information for associated tasks.

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

## A  CONSTRUCTING SYNTH-TS

**Gaussian grid.**  We first construct an $N_\mu \times N_\mu$ grid to define the means of the Gaussians from which we sample $(f, k)$. For simplicity, we assume the two dimensions of the Gaussians are independent, with standard deviations $\sigma_f$ and $\sigma_k$ for all Gaussians.

The mean frequency and slopes are defined as follows:

$$\mu_f \in [\mu_0, \mu_0 + 2\sqrt{2}\sigma_f, \cdots, \mu_0 + 2(N_\mu - 1)\sqrt{2}\sigma_f], \tag{17}$$

$$\mu_k \in [-\sqrt{2}N_\mu\sigma_k, (-N_\mu + 2)\sqrt{2}\sigma_k, \cdots, (-N_\mu + 2(N_\mu - 1))\sqrt{2}\sigma_k]. \tag{18}$$

In the frequency direction, the distance between two points sampled from the same Gaussian follows a normal distribution $\mathcal{N}(0, 2\sigma_f^2)$. Conversely, for two points sampled from distinct Gaussians, their distance follows a distribution $\mathcal{N}(2\sqrt{2}k\sigma_f, 2\sigma_f^2)$ for $k \geq 1$. The results for the slope direction are analogous. In our experiments, we set $\mu_0 = 0.2$, $\sigma_f = 0.4$, and $\sigma_k = 0.5$.

**Seasonal component.**  From each Gaussian in the grid, we generate time series of duration 2 seconds and sampling rate 256Hz. The sine waves and triangle waves are defined as follows:

$$\text{Sine}(f, t) = |\sin(\pi f t)| + b, \tag{19}$$

$$\text{Triangle}(f, t) = \begin{cases} 4fx - 1 + b & 0 \leq x < \frac{1}{2}, \\ -4fx + 2 + b & \frac{1}{2} \leq x < 1, \end{cases} \tag{20}$$

where $x = tf - \lfloor tf \rfloor$ and the offset $b$ is set to 1.

**Trend component.**  The trend component, parameterized by $k$, is defined as follows:

$$\text{Trend}(k, t) = \begin{cases} kt & k \leq 0, \\ k(t - \max(t)) & k < 0. \end{cases} \tag{21}$$

**Gaussian noise.**  For each sampled data point on the time series, we added a Gaussian noise component: $Noise \sim \mathcal{N}(0, \sigma_n^2)$. We set $\sigma_n = 0.02$ in our experiments.

## B  AMR PROFILE PREPROCESSING IN DRIAMS

AMR profiles The DRIAMS dataset uses a mixed labeling scheme to represent AMR. Each AMR label can be one of 'S'(susceptible), 'R'(resistant), 'I'(Intermediate), '1'(susceptible or intermediate), '0'(susceptible), or unknown (we labeled as 'N').

We first build the following similarity lookup table between labels:

|   | S | I | R | 1 | 0 | N |
|---|---|---|---|---|---|---|
| S | 1 | 0 | 0 | 0 | 1 | 0 |
| I | 0 | 1 | 0 | 1 | 0 | 0 |
| R | 0 | 0 | 1 | 1 | 0 | 0 |
| 1 | 0 | 1 | 1 | 1 | 0 | 0 |
| 0 | 1 | 0 | 0 | 0 | 1 | 0 |
| N | 0 | 0 | 0 | 0 | 0 | 0 |

Table 2: Similarity between AMR labels in DRIAMS.

For a pair of AMR profiles, we look up and sum the similarity for each pair of AMR labels to get the similarity between the AMR profiles. For example, profiles ['1', 'S', 'N'] and ['I', '0', 'R'] will have similarity $1 + 1 + 0 = 2$. The *dissimilarity* between the two profiles is calculated by subtracting the similarity from the length of the AMR profiles, i.e., the number of drugs.

## C  SPECIES LABEL DESCRIPTIONS OF THE PROPRIETARY DATASET

| Label | Species |
|-------|---------|
| ACIN | Acinetobacter baumannii |
| BC | Burkholderia cepaciae |
| CB | Citrobacter |
| EB | Enterobacter cloacae |
| EC | Escherichia coli |
| KLO | Klebsiella oxytoca |
| KLP | Klebsiella pneumoniae |
| MRSA | Staphylococcus aureus |
| PR | Proteus mirabilis |
| PRV | Providencia |
| PSA | Pseudomonas aeruginosa |
| PSB | Pseudomonas |
| SER | Serratia marcescens |
| STEN | Stenotrophomonas maltophilia |
| VRE | Vancomycin-resistant Enterococcus |

Table 3: Reference table for species appearing in Figure 5.

## D  LIFT IN F1 SCORE FOR MSPL VS ONLYCLS ON DRIAMS

**(a)**                                        **(b)**

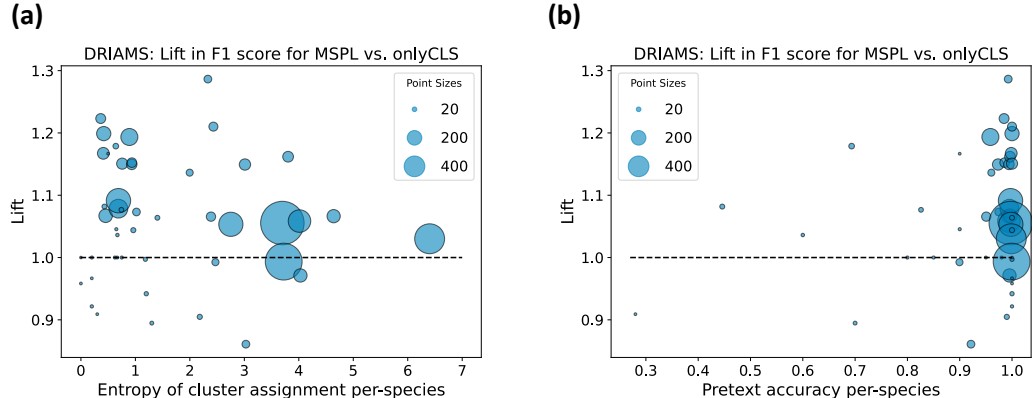

Figure 7: Lift in F1 score between MSPL and onlyCLS for different species on DRIAMS, sorted by the entropy of ground truth clusters or the pretext accuracy. The dot size reflects the number of MALDI samples in that species.

