# OpenReview forum: "Multimodal Structure Preservation Learning"
_ICLR.cc/2025/Conference — Submitted to ICLR 2025_

### Official Review · Reviewer_eGyy · 2024-11-01

**Soundness:** 3
**Presentation:** 2
**Contribution:** 2
**Rating:** 3
**Confidence:** 3

**Summary:**

The authors propose the Multimodal Structure Preservation Learning (MSPL) approach that learns data representations utilizing clustering structure in one data modality to inform upon the other modality using a regularization approach towards compliance of this clustering structure when learning representations. The approach is applied to synthetic as well as whole genome sequencing (WGS) and antimicrobial resistance datasets. Rather than learning a shared feature space the approach thus relies on gross structural information at the level of groups exploring alignment according to the dissimilarity-based clustering learned by the opposing modality. The approach relies on three tasks, an autoencoder for learning representations, a pretext discriminatory task, and  alignment of the two modalities clustering structure formulated as a multiobjective function reflected in three loss terms with associated relative weights. Apart from conventional ARI and NMI cluster validity metrics the authors further propose a cluster-based F1 score. The approach is compared against two model ablations (baselines) not having the structure preserving loss and classifying the cluster groups respectively as opposed to operating on dissimilarities.

**Strengths:**

The approach is useful and enable to integrate information of multiple (two) modalities taking overall structural information into account from the opposing modality.

The considered problem domain is interesting and the approach’s seem to enhance the learned representations in terms of cluster level structures.

The paper is well written and easy to follow.

**Weaknesses:**

The methodological contribution of the paper is very limited and rather straightforward combining three loss components. As such, the contribution seems rather incremental and limited in scope.

The contribution of the F1 metric is also straightforward and does not contribute much in terms of novelty.

The comparisons are very limited only considering simple model ablations but not any alternative state-of-the-art methodology for the same problem domain.

The results are not overly convincing with the approach working better than baselines in some situations and not in other.

Overall I find the contribution of limited novelty and the experimentation not overly convincing - and therefore do not recommend publication at this point.

**Questions:**

It would be good to further discuss how to suitable tune the contribution of each loss term.

How is the approach influenced by initialization conditions?

How does architectural choices influence the model and why is UNETs chosen as the backbone as opposed to other architectures such as transformer based architectures?

Why is the approach not compared to any existing SOTA approaches within the domain or similar domains for instance based on the approaches reviewed in related work?

The results are also not that surprising in that regularizing towards a clustering structure will enhance such learning of the clustering structure. It would in this context be interesting to see if the regularization also improves upon the pretext class and contrast this to other methodologies directly learning the pretext class.

---

> ### Author Response · Authors · 2024-11-27
> **Response to the Reviewer**
>
> Thank you for your time and help. We respond to your concerns as follows:
>
> **Weaknesses: The methodological contribution of the paper is very limited and rather straightforward combining three loss components. As such, the contribution seems rather incremental and limited in scope.
> The contribution of the F1 metric is also straightforward and does not contribute much in terms of novelty.
> The comparisons are very limited only considering simple model ablations but not any alternative state-of-the-art methodology for the same problem domain.
> The results are not overly convincing with the approach working better than baselines in some situations and not in other.
> Overall I find the contribution of limited novelty and the experimentation not overly convincing - and therefore do not recommend publication at this point.**
>
> The novelty of our approach is two-fold:
> 1. There has been no SOTA method in solving the MSPL problem with one raw data modality and another distance-based data modality. To our knowledge, our approach is the first such attempt at the problem.
> 2. Our MSPL framework is of practical value in outbreak detection: If our method works, the hospitals can forgo WGS sequencing and use the more cost-effective, more generally accessible MALDI, democratizing access to the in-hospital outbreak detection that can save lives.
>
> **Q1. It would be good to further discuss how to suitable tune the contribution of each loss term.
> How is the approach influenced by initialization conditions?**
>
> **A1.** Thank you for these remarks. We will investigate these two questions in a future manuscript.
>
> **Q2. How does architectural choices influence the model and why is UNETs chosen as the backbone as opposed to other architectures such as transformer based architectures?**
>
> **A2.** Convolution-based UNET is chosen because it is lightweight and is widely adapted in mass-spectrometry-related tasks: see https://www.nature.com/articles/s41540-024-00385-x and https://www.sciencedirect.com/science/article/abs/pii/S0167701221002463#s0010
>
> **Q3. Why is the approach not compared to any existing SOTA approaches within the domain or similar domains for instance based on the approaches reviewed in related work?**
>
> **A3.** The approaches listed in the related work sections are related but not directly applicable to our problem.
>
> **Q4. The results are also not that surprising in that regularizing towards a clustering structure will enhance such learning of the clustering structure. It would in this context be interesting to see if the regularization also improves upon the pretext class and contrast this to other methodologies directly learning the pretext class.**
>
> **A4.** Thank you for this valuable insight. We will investigate the effect of regularization on the pretext task, though this is not the main objective of the current paper.

---

> ### Comment · Reviewer_eGyy · 2024-11-29
> **Rebuttal insufficiently address my concerns and I am inclined to maintain my score**
>
> I thank the authors for their response but find that they insufficiently address my concerns leaving key aspects as future work or something they will investigate but without producing the requested results at this point.

---

### Official Review · Reviewer_5YWd · 2024-11-03

**Soundness:** 2
**Presentation:** 2
**Contribution:** 2
**Rating:** 3
**Confidence:** 5

**Summary:**

The paper proposed a domain adaptation method that learns the data distribution structure in one modal and transfer it to the other modal. They applied it to the problem of hospital outbreak detection that using MALDI and whole genome sequencing.

**Strengths:**

The stated problem is pervasive in biomedical applications and is challenging.

**Weaknesses:**

1) This is a typical subset of domain adaptation problems. However, they did not include SOTA domain adaptation methods into the baseline. The baseline methods are weak.
2) Also, from references we see that there are already methods that perform prediction tasks directly based on MALDI, which were not compared.
3) The experiments are carried out only on MALDI-WGS datasets and most are synthetic datasets. Due to the small-sample nature of these problems, the models are vulnerable to short-cut learning and testing on several similar datasets is not reliable. I don't see any reason that the problem should be restricted on MALDI-WGS data. There are lots of two-domain problems with similar character in biomedical fields and the data should be tested on more types of applications.

**Questions:**

How does the "seasonal and trend components" in the synthetic datasets related to MALDI-WGS matching?

---

> ### Author Response · Authors · 2024-11-27
> **Response to the Reviewer**
>
> Thank you for your time and help. We respond to your concerns as follows:
>
> **W1. This is a typical subset of domain adaptation problems. However, they did not include SOTA domain adaptation methods into the baseline. The baseline methods are weak.**
>
> We beg to disagree. Our view of the problem presented in this paper is that it is a representation learning problem involving not only different domains of data but also different types of representations. We intend to show how to leverage one to enhance the results attainable with the other, removing dependency on the first domain from the application after the model was trained. So effectively, we aim to replace WGS-based outbreak detection with MALDI-based outbreak detection, but leveraging the knowledge embedded in the WGS data to enhance the MALDI-based approach.
>
> **W2. Also, from references we see that there are already methods that perform prediction tasks directly based on MALDI, which were not compared.**
>
> To our knowledge, there are no existing methods that predict WGS cluster labels from MALDI. If we compare our approach with other methods by modifying their prediction objectives, they may not be adequate for baseline comparison).
>
> **W3. The experiments are carried out only on MALDI-WGS datasets and most are synthetic datasets. Due to the small-sample nature of these problems, the models are vulnerable to short-cut learning and testing on several similar datasets is not reliable. I don't see any reason that the problem should be restricted on MALDI-WGS data. There are lots of two-domain problems with similar character in biomedical fields and the data should be tested on more types of applications.**
>
> We agree that the method should eventually be tested on more applications besides MALDI-WGS. It is our primary application focus as of now, though. We will look to include other application scenarios in the future.
>
> **Q1. How does the "seasonal and trend components" in the synthetic datasets related to MALDI-WGS matching?**
>
> They are not directly related to each other. In the synthetic datasets, the input data are time series with seasonal & trend & noise components; the pretext task is the classification of the seasonal component; the MSPL objective is to infer the finer-grained information of what Gaussian the frequency of the seasonal component is sampled from. In MALDI-WGS, the pretext task is MALDI species identification and the MSPL objective is to recover the WGS/SNP-defined clusters.
>
> Again, we thank you for your insightful comments.

---

> > ### Comment · Reviewer_5YWd · 2024-11-29
> >
> > Thanks the authors for taking time to response. However, if the paper's scope is limited to ad-hoc solution for MALDI-WGS matching, it's quite narrow and unfit for ICLR.

---

### Official Review · Reviewer_2upn · 2024-11-03

**Soundness:** 1
**Presentation:** 3
**Contribution:** 1
**Rating:** 1
**Confidence:** 4

**Summary:**

This paper presents an approach to multimodal representation learning that leverages an autoencoder along with a combination of 3 loss functions (reconstruction, pretext task performance and structural alignment) in order to learn representations of the data that preserve the structure in one of the modalities (represented by a dissimilarity matrix) without requiring the raw data itself.

The authors apply the method in the context of epidemiology, where mass spectrometry data is becoming a potentially valuable tool for outbreak detection but is limited in power compared to whole genome sequencing, which can be a prohibitively labor-intensive and costly approach. They present the method as a way of integrating these modalities. The method is evaluated on a simulated dataset, a public dataset of paired MALDI spectra and antibiotic resistance data, and a proprietary dataset of WGS structural data and MALDI spectra. To evaluate, the authors compare their proposed method (MSPL) to two baseline methods that they construct without the structure alignment loss function, and evaluate clusterings based on the resulting representations using a variety of extrinsic clustering metrics with respect to ground truth.

**Strengths:**

The concept of preserving structure level alignment without need for the entire dataset is interesting, and the proposed approach appears to be novel. The application of multimodal deep representation learning approaches of this kind to mass spectrometry data in the context of epidemiology is particularly original and exciting.

The method is very clearly described, as is the evaluation approach and the metrics used. In evaluating, the authors considered extrinsic clustering metrics that went beyond more common approaches such as ARI and NMI, which greatly assist in the interpretation of the results.

Additionally, the authors evaluate the method on multiple datasets, including a variety of simulations and two real-world datasets, which are well-described.

**Weaknesses:**

The paper has several significant weaknesses.

First, the significance of the method’s real-world impact in the application area is somewhat unclear. The introduction states that the main utility of the learned representation in this context is that it could replace WGS in practice as a more cost-effective alternative; however, the method seems to require SNP distances between each pair of samples (and thus WGS for every sample) as an input in order to learn the representation. As such, it is not clear how such representations would be learned without doing WGS first – thus incurring the same costs as would be necessary to do outbreak detection in the usual way. This somewhat reduces the perceived contribution of the work.

The evaluation approach is also a major weakness of the paper. The performance of the model is poor in many cases, and the proposed metrics make it very difficult to understand why. Cluster purity, precision, recall and F1 scores for clustering have already been defined in existing literature  – see the chapter on “Evaluation of Clustering” in Information Retrieval by Manning. In order to deal with the challenge of comparing clusters of different sizes and number, precision, recall and F1 score are typically defined with respect to the cluster memberships of sample pairs. However, the paper defines these metrics very differently: with respect to purity, which is easy to achieve when cluster sizes are large, and makes the results very difficult to interpret. For example, while the F1 scores seem generally high, they appear to be driven predominantly by a sharp increase in recall. Figure 6 demonstrates that MSPL learns many fewer clusters than the ground truth – if MSPL is also learning fewer or larger clusters than the baseline models, then this could easily explain the increase in the purity-based recall metric. Although the purity-based precision metric decreases in these cases, it could also be artificially inflated or otherwise biased by cluster size or distribution. Unfortunately, the number of clusters learned in each experiment is not reported, which makes evaluation even more difficult. The NMI and ARI metrics are designed to account for these potential sources of bias, but the authors were not able to demonstrate that MSPL consistently outperforms the baselines according to these metrics in real-world data. Overall, the evaluation approach should be reformulated to be consistent with the literature and the results require much more investigation.

The choice of baselines is also a substantially limiting factor. While the authors construct two baselines, the paper does not make any comparison of MSPL to existing methods. While relevant deep learning approaches may be limited, there are many papers on late integration multi-view clustering approaches, which integrate multiple modalities using only clustering labels or dissimilarity matrices and not the original data (see, e.g. “Multi-omic and multi-view clustering algorithms: review and cancer benchmark” by Rappaport et al for a brief review of such approaches). Since the evaluation in the paper is based entirely on the quality of clustering based on the learned representation, this class of methods seem very relevant. Furthermore, there is no attempt to evaluate how well the model performs in comparison to models that leverage the entire dataset rather than just the distance-based structure. This makes it difficult to evaluate the real-world utility of the method.

Finally, there is no reproducibility statement and no mention of code or data being made available.

**Questions:**

The above comments raise some broader questions regarding the method and particularly the evaluation approach. Some more specific questions are listed below:

- Regarding the choice of the custom loss function for SNP distance, the choice to impose no penalty when feature distance and SNP distance both exceed 15 seems confusing. The cited source suggests that a distance <= 5 indicates a definite transmission  <= 15 indicates probable transmission. When clustering this data, might it also be useful to have a representation that accurately captures the relationships between samples that are even somewhat less likely to cluster together? If so, it could make sense to either relax this constraint or try a different normalization approach (such as log transforming the SNP distances).

- The model requires the choice of a pretext task, and the authors suggest that the difficulty of the pretext task does not affect MSPL’s ability to preserve structure. What, then, is the effect of the pretext task on the learned representation, and how should a user choose the pretext task for their particular application?

Some minor comments on Figure 5 that did not affect my score:
-  e) and f) are missing species labels.
-  The paper claims that F1 lift and species diversity are correlated based on c) and e) – a regression line or correlation statistic would be helpful to back up this claim.

---

> ### Author Response · Authors · 2024-11-27
> **Response to Weaknesses**
>
> Thank you for your time and help. We respond to your concerns ("**Weaknesses**") as follows:
>
> **W1. First, the significance of the method’s real-world impact in the application area is somewhat unclear. ... This somewhat reduces the perceived contribution of the work.**
>
> **A1.** To train our model, we use MALDI + SNP distance in hopes of imbuing the structure of SNP into the representations of MALDI. During evaluation/inference, the model has no knowledge of the SNP for the data provided as queries. Hence, in an epidemiology application scenario, a previously trained model fed with only MALDI data as input, is expected to produce representations and predictions whose pairwise distances mimic SNP distances WITHOUT the need to conduct WGS sequencing.
>
> **W2. The evaluation approach is also a major weakness of the paper. ... Overall, the evaluation approach should be reformulated to be consistent with the literature and the results require much more investigation.**
>
> **A2.** We are motivated to rely on the F1 score by the epidemiologists, who believe that isolates clustered together w.r.t.  the ground-truth SNP distance should also be clustered together in MALDI data space — this desired matching can be  reflected by recall, which we replaced by the F1 score to avoid trivial optima.
> We agree that we need to further investigate the behavior of MSPL in terms of the number of predicted clusters in order to be consistent with the literature.
>
> **W3. The choice of baselines is also a substantially limiting factor. ... This makes it difficult to evaluate the real-world utility of the method.**
>
> **A3.** Thank you very much for sharing the reference. Our problem is different from multi-omic/multi-view clustering: the latter is about achieving unified clustering results by aggregating information from multiple sources. However, our problem amounts to learning a clustering of one modality supervised by the clustering of another, i.e., the two modalities do not jointly produce a “new” clustering assignment. Nevertheless, we believe that multiview clustering of MALDI+SNP would be worth investigating as a separate research thread.
> In the MALDI-SNP scenario, we are only given the SNP distance structure and not the raw sequencing data. We believe that structure preservation can be better achieved with raw sequencing data, given the development of DNA foundation models. It is this practical constraint that motivated us to develop the MSPL framework.
>
> **W4. Finally, there is no reproducibility statement and no mention of code or data being made available.**
>
> Thank you for this note. We will release the code and data upon paper acceptance. We indeed should have made it clear in the original submission.

---

> ### Author Response · Authors · 2024-11-27
> **Response to Questions**
>
> Thank you for your time and help. We respond to your concerns ("Questions") as follows:
>
> **Q1. Regarding the choice of the custom loss function for SNP distance, the choice to impose no penalty when feature distance and SNP distance both exceed 15 seems confusing ... If so, it could make sense to either relax this constraint or try a different normalization approach (such as log transforming the SNP distances).**
>
> **A1.** In the particular application to outbreak detection, all SNP distances greater than 15 are treated as negative, i.e., the involved isolates are assumed unrelated. The loss reflects this viewpoint in that we only require predictions for >15 ground truth SNP to also be >15, but do not insist on high accuracy of SNP reconstruction in that range of distances, focusing the loss function on accurately reflecting the closer matches. We also tried to log-transform the SNP distances, but this resulted in poor performance against data of the key interest in SNP-based similarity.
>
> **Q2. The model requires the choice of a pretext task, and the authors suggest that the difficulty of the pretext task does not affect MSPL’s ability to preserve structure. What, then, is the effect of the pretext task on the learned representation, and how should a user choose the pretext task for their particular application?**
>
> **A2.** Our approach aims at extending and quantifying the utility of MALDI to outbreak detection, which is typically conducted via WGS. The pretext task of species identification is the original utility of MALDI. In other words, we learn representations of MALDI that can be both used in species identification and outbreak detection.
> In investigating the effect of the pretext task on MSPL, we do not vary our pretext task: pertaining to a fixed model, we have samples of MALDI data whose species are either hard or easy to classify; we simply examine the species-wise clustering performance. The reason for this investigation is apparent — given that outbreak detection operates at the sub-species strain level, we want to know if the model struggles to recover WGS clusters, and if it does, is it because the model struggles to learn the coarser species-level information? Our conclusions are negative, meaning that although there is some hierarchy between the two tasks, the final performance does not appear to be entangled with such a relationship.
> Nevertheless, the choice of the pretext task for MALDI is relevantly straightforward since species identification is the most common use of MALDI. For other data modalities, the choice of pretext task may be non-trivial.
>
> **Q3. Some minor comments on Figure 5 that did not affect my score:
> e) and f) are missing species labels.
> The paper claims that F1 lift and species diversity are correlated based on c) and e) – a regression line or correlation statistic would be helpful to back up this claim.**
>
> **A3.** Thank you very much for the comments on Figure 5. We will modify the Figure accordingly.

---

> ### Comment · Reviewer_2upn · 2024-12-03
> **Thank you for the authors' answers; some major weaknesses not sufficiently addressed**
>
> Thank you to the authors for the time they took to address these concerns and questions. In particular, the method's value for inference on data that does not include WGS sequencing is notable and much appreciated. The explanation of the pretext task analysis is also helpful, although the impact of the choice of pretext task on performance could still be clarified.
>
> The evaluation approach and the choice of baselines remain significant weaknesses of the submission and require substantial additional work to address. Unfortunately, these issues remain a barrier to publication.

---

### Official Review · Reviewer_tU7S · 2024-11-04

**Soundness:** 2
**Presentation:** 2
**Contribution:** 1
**Rating:** 3
**Confidence:** 4

**Summary:**

This paper presents a multimodal framework called MSPL, which builds upon encoder decoder structure with extra regularizations form prediction task and structure loss.

**Strengths:**

I think the paper has several strenghs:
1: It presents a flexible framework that can incorporate different modality as inputs, incorporating various loss functions and clustering objectives
2: It addresses a real-world problem in epidemiology (using MALDI data as a cost-effective alternative to WGS)
3: It introduces a new cluster evaluation metric (cluster F1 score)

**Weaknesses:**

This paper has several areas that can be improved:
1: lt could benefit from more extensive comparison with other multimodal learning approaches
2: Authors could explore more sophisticated structure preservation objectives The three losses are common objective functions in multimodal and VE/VAE variants. Besides, there is limited discussion of the impact of different encoder architectures
3: Model needs further optimization. Even comparing with its own variants, the proposed model cannot outperform them in most cases.
4: I am not sure if it can handle a large number of clusters or clusters with imbalanced sizes.

**Questions:**

See above.

---

> ### Author Response · Authors · 2024-11-27
> **Response to the Reviewer**
>
> Thank you for your time and help. We respond to your concerns as follows:
>
> **1. lt could benefit from more extensive comparison with other multimodal learning approaches.**
>
> Other multimodal learning approaches commonly require paired raw data. We focus on the setting where we only have raw data for one modality and pairwise distances (between data points) for another. Hence, other multimodal learning approaches may not be directly comparable. Our work indeed aims to extend multimodal learning to such settings where we admit multi-modality of data representations, not just multi-modality of sources or views.
>
> **2: Authors could explore more sophisticated structure preservation objectives. The three losses are common objective functions in multimodal and VE/VAE variants. Besides, there is limited discussion of the impact of different encoder architectures.**
>
> We agree that the structure preservation objective can be more sophisticated. We will explore more complex objectives and encoder architectures in our future work and include this note in the conclusions.
>
> **3: Model needs further optimization. Even comparing with its own variants, the proposed model cannot outperform them in most cases.**
>
> As demonstrated in Table 1, in almost all datasets, the current model outperforms all other methods in the F1 Score. Also, when constraining the number of output clusters to match that of the ground truth, our method results in a superior NMI score.
> Nevertheless, as described in the response to 2, we will optimize the encoder architecture and structure preservation objectives.
>
> **4: I am not sure if it can handle a large number of clusters or clusters with imbalanced sizes.**
>
> As described in the Limitations paragraph in the Discussion section, we concur that our method is currently limited in handling imbalanced cluster distributions. This is a direction that we will try to improve on in the future.
>
> Again, we thank you for your insightful comments.

---

### Meta-Review · Area_Chair_mUxD · 2024-12-13

**Metareview:**

This paper proposes a multimodal representation learning method called MSPL, which leverages an autoencoder with three loss functions: reconstruction, pretext task performance, and structural alignment. MSPL learns to preserve the structure of one modality, represented as a dissimilarity matrix, without requiring raw data. The method is applied to hospital outbreak detection using MALDI spectra and whole genome sequencing (WGS) data, addressing the limitations of WGS in cost and labor. It is evaluated on simulated, public, and proprietary datasets, comparing MSPL to baselines without structural alignment loss, using extrinsic clustering metrics to assess performance.

As the reviewers pointed out, there is significant room for improvement in the paper, including hyperparameter tuning and comparison to existing methods. Therefore, it is difficult to accept the paper in its current form. I encourage the authors to revise the paper based on the reviewers' comments and resubmit it to a future venue.

**Additional Comments On Reviewer Discussion:**

Some concerns are addressed during the rebuttal period. However, the main concern (comparison to existing methods) was not clearly addressed.

---

### Decision · Program_Chairs · 2025-01-22

Reject